# Exploring Neural Mechanisms of Reward Processing Using Coupled Matrix Tensor Factorization: A Simultaneous EEG–fMRI Investigation

**DOI:** 10.3390/brainsci13030485

**Published:** 2023-03-13

**Authors:** Yuchao Liu, Yin Zhang, Zhongyi Jiang, Wanzeng Kong, Ling Zou

**Affiliations:** 1School of Computer and Artificial Intelligence, Changzhou University, Changzhou 213164, China; 2School of Microelectronics and Control Engineering, Changzhou University, Changzhou 213164, China; 3College of Computer Science, Hangzhou Dianzi University, Hangzhou 310018, China; 4Key Laboratory of Brain Machine Collaborative Intelligence Foundation of Zhejiang Province, Hangzhou 310018, China

**Keywords:** simultaneous EEG–fMRI, reward processing, tensor factorization, data fusion, blind source separation

## Abstract

Background: It is crucial to understand the neural feedback mechanisms and the cognitive decision-making of the brain during the processing of rewards. Here, we report the first attempt for a simultaneous electroencephalography (EEG)–functional magnetic resonance imaging (fMRI) study in a gambling task by utilizing tensor decomposition. Methods: First, the single-subject EEG data are represented as a third-order spectrogram tensor to extract frequency features. Next, the EEG and fMRI data are jointly decomposed into a superposition of multiple sources characterized by space-time-frequency profiles using coupled matrix tensor factorization (CMTF). Finally, graph-structured clustering is used to select the most appropriate model according to four quantitative indices. Results: The results clearly show that not only are the regions of interest (ROIs) found in other literature activated, but also the olfactory cortex and fusiform gyrus which are usually ignored. It is found that regions including the orbitofrontal cortex and insula are activated for both winning and losing stimuli. Meanwhile, regions such as the superior orbital frontal gyrus and anterior cingulate cortex are activated upon winning stimuli, whereas the inferior frontal gyrus, cingulate cortex, and medial superior frontal gyrus are activated upon losing stimuli. Conclusion: This work sheds light on the reward-processing progress, provides a deeper understanding of brain function, and opens a new avenue in the investigation of neurovascular coupling via CMTF.

## 1. Introduction

Reward processing is central to emotional decision-making and risk-taking [1]. It is often used in mental health diagnoses as many psychiatric disorders are accompanied by certain deficiencies while processing rewards [2,3]. Yet, comprehending how the brain processes rewards still remains a non-trivial outstanding question. A typical reward-processing progress is made up of two parts: the anticipation and the outcome phase, each of which involves several subprocesses that co-occur on different time scales. Reward anticipation usually involves the cognitive operations that precede the arrival of a reward, including incentive evaluation, probability estimation, and motor preparation. On the other hand, reward outcome corresponds to those operations triggered by the actual delivery of a reward, such as hedonic feelings, reward value updating, and behavioral reinforcement [4]. The gambling task is a well-established classic paradigm suitable for studying the cognitive processes of reward processing, which are usually investigated by EEG and fMRI. The EEG data analysis of reward processing often relies on event-related potentials (ERPs), time-frequency analysis, and blind source separation (BSS) [5]. Foti et al. [6] used the EEG to track ERPs associated with reward processing. Glazer et al. [7] studied ERPs associated with reward processing as well as EEG time-frequency components. Sambrook et al. [8] used BSS to isolate most sources of ongoing brain activity without a priori knowledge about source signals or mixing processes. An analysis of fMRI data for reward processing generally employs the general linear model (GLM). Jauhar et al. [9] used the GLM to obtain brain regions associated with reward processing from fMRI data.

Studies have found that various brain regions and networks are involved in these processes. The main network here is the so-called reward system, which includes the nucleus accumbens, ventral tegmental area, substantia nigra, amygdala, the basal forebrain, and prefrontal cortex. Moreover, activity in the hippocampus and parahippocampal gyrus seems to be associated with punishment. There are also brain areas that react to arousal during risk-taking behavior, regardless of whether the outcome is positive or negative, including the orbitofrontal cortex, insula, and head of the caudate [10]. Similarly, the ventral striatum, cingulate cortex, and insula show activation in response to reward [9]. Furthermore, the reward degree during reward processing positively correlates with brain activation in the bilateral striatum [11]. Because of this intricacy, it is necessary to combine complementary neuroimaging techniques to understand how the brain processes the rewards fully.

The EEG and fMRI are two of the most popular non-invasive brain imaging techniques used in cognitive science that help describe underlying brain networks and neurobiological processes. The EEG can reveal fast brain dynamics due to its excellent temporal resolution (millisecond level). However, its spatial resolution is low. In contrast, the fMRI can reflect precise functional changes in brain activity due to its satisfactory spatial resolution (millimeter level), while the temporal resolution is very poor. It is therefore of vital importance to find a way to fuse complementary different brain imaging techniques [12,13], such as the emerging field of deep learning-based fusion [14]. Currently, there are three main EEG–fMRI fusion methods: EEG source localization based on fMRI constraints [15,16]; fMRI prediction based on EEG information [17]; and symmetric fusion methods that simultaneously interpret data from both modalities [18]. Among these, the symmetric fusion methods are mainly used to separate the sources of EEG and fMRI data using joint decomposition methods such as ICA and CCA [19]. Such techniques have already benefited numerous branches of cognitive neuroscience, including the prediction and localization of epileptic lesions [20] and coupling mechanisms between EEG and fMRI [21]. Dehghani et al. [22] implemented EEG neurofeedback with simultaneous fMRI to understand the effect of neurofeedback on brain activity and the interaction of whole brain regions involved in emotion regulation. To date, only a research team has investigated reward processing using simultaneous EEG–fMRI fusion techniques and obtained more accurate results than using either the EEG or fMRI [23].

BSS methods that use matrices as input, such as ICA, can separate multivariate signals into additive components based on the assumption that they are statistically independent. However, this approach is deficient in exploiting the inherent multiway nature of these data [24]. Studies in only two dimensions, temporal and spatial, cannot reveal the inherent higher-order structure of the data nor the interdependence of the different dimensions. EEG and fMRI data are inherently multidimensional, which means they include information about time, different voxels or channels, subjects, trials, etc. For the EEG, the signal can be expanded along additional patterns in order to reveal more potential information. This multidimensional nature of EEG and fMRI data suggests the use of tensor models rather than matrix models. 

Tensor decomposition models can improve the ability to extract spatiotemporal patterns of interest, facilitate neurophysiologically meaningful interpretations, and produce unique representations under mild conditions [25]. In addition, tensor methods are able to make predictions more robustly in the presence of noise than matrix-based methods [26]. Due to the superiority of tensor analysis methods on multidimensional neuroimaging data, the CMTF was proposed [27], which is a unique coupled BSS method. The BSS method views EEG and fMRI data as a superposition of various physiological and non-physiological sources [28]. It is inherently adapted to higher-order data presented as a tensor, which can capture the rich structure in the data. On this basis, the CMTF can also estimate the shared components between the two modal data and perform symmetric processing of EEG and fMRI to achieve a true fusion.

The CMTF method has been applied in several EEG and fMRI fusion areas. Van Eyndhoven et al. [29] used the CMTF to make inferences on the localization of the ictal onset zone in refractory focal epilepsy based on simultaneous EEG and fMRI recordings. Mosayebi et al. [30] applied the CMTF to a real dataset of an auditory oddball paradigm and found that the CMTF method had better results and higher performance than the N-way partial least square (N-PLS) method. Hunyadi et al. [31] used the CMTF method for epilepsy studies to explore epileptic network activity and obtained more stable results than jointICA. Rivet et al. [32] used the CMTF method to study the ocular artifacts in EEG data. Acar et al. [33] used the CMTF method to capture the difference between schizophrenic patients and healthy controls in brain activity patterns. Mosayebi et al. [34] employed the CCMTF method for an analysis of the emotion regulation paradigm.

Since many matrix and tensor decompositions must be solved using non-convex optimization-based algorithms, these algorithms may converge to a local optimum. Therefore, the decomposition needs to be computed with multiple initializations to verify that the decomposed components are reproducible in the optimization. Clustering algorithms are needed to cluster the results of multiple decompositions to assess the reliability or stability of the decomposition.

In this paper, the unknown neurovascular coupling between electrophysiological phenomena, measured by EEG, and hemodynamic changes, captured by blood oxygen level-dependent (BOLD) signals, is explained using the hemodynamic response function (HRF). EEG data were expanded into a tensor (time points × frequencies × channels) and then decomposed along a common temporal dimension with fMRI matrix data using CMTF. After using graph-structured clustering, the best model was selected by four quantitative metrics: CORCONDIA, reproducibility, similarity, and significance. Neural imaging data from multiple modalities were jointly decomposed as a superposition of multiple sources to extract shared information and distinguish between similarities and differences in modalities. The simultaneous EEG-fMRI data of reward-processing decomposition results were interpreted in four dimensions: time, frequency, channel, and region of interest (ROI). This is the first attempt to apply tensor decomposition to simultaneous EEG and fMRI data in a gambling task. The fact that the CMTF method can reveal the inherent higher-order structure of the data and the interdependence of the different dimensions makes it possible to discover additional ROIs which were previously ignored in other literature during the processing of rewards.

## 2. Materials and Methods

Using the CMTF method, the simultaneous EEG-fMRI data were jointly decomposed to capture ROIs activated by reward processing. The collected data then underwent a sequence of preprocessing, dimensional expansion of EEG data, CMTF, and model selection. After using graph-structured clustering, the best model was selected by four quantitative metrics, CORCONDIA, reproducibility, similarity, and significance, to obtain the fMRI components (ROI factors) and the corresponding EEG components (frequency factors, channel factors, and time factors). Figure 1 depicts the complete process from data preprocessing to the results.

### 2.1. Subjects

The following inclusion criteria were applied: (1) 18 to 25 years of age; (2) normal or corrected normal vision; (3) right-handed. The following exclusion criteria were applied: (1) use of corticosteroids and psychotic drugs within the past 30 days; (2) current or previous history of neurological, medical, or psychiatric disorders; (3) current or previous history of neurosurgery, head injury, cerebrovascular injury, or traumatic brain injury involving loss of consciousness; (4) presence of learning disability; (5) presence of claustrophobia; (6) refusal to give informed consent; and (7) presence of magnetic implants in the body.

Twenty healthy, right-handed subjects with an average age of 23 years (17 men and 3 women; range 19−25 years; SD ± 1.48) were involved in this study; we previously studied them in [23]. Subjects had normal or corrected normal vision without a history of neurological, medical, or psychiatric disorders. The ethics committee (Changzhou University, Changzhou, China) approved the experiment, and all subjects signed an informed consent form before the experiment. They received comprehensive instructions about the gambling task. Following the examination, each subject was interviewed to ensure the task was handled appropriately and to learn more about the gambling strategy applied. Subjects were asked to share their thoughts and theories about the experimental setup. Incorrect handling of the paradigm, for example, consistently pressing buttons at the wrong time or misinterpreting the task, could lead to subject exclusion, but this never occurred for the tested subjects. The subjects’ complete clinical data are shown in Table 1.

### 2.2. Gambling Task

A classic gambling task [35] was designed using E-prime 3.0 software (Psychology Software Tools, Pittsburgh, PA, USA). The experimental procedure consisted of 8 recurrent tasks, each containing 10 trials, for a total of 80 trials, consisting of random presentations of winning and losing stimuli. A single-trial flow is shown in Figure 2. Two identical doors would appear simultaneously on the screen, one corresponding to a win (+USD 2.0) and the other to a loss (−USD 1.0). Subjects were instructed beforehand to always choose the door that they thought would lead to winning the money using an MRI-compatible response box. The computer would randomly select a door if the subject did not make a timely choice. A 2000 ms gaze point and a 2000 ms feedback arrow came afterward. A green up arrow indicated a win, and a red down arrow a loss. At the end of the arrow feedback, the center of the screen would display the subject’s current cumulative score. After each trial, subjects were given a 4000 ms break. Before the formal experiment, subjects were required to complete two exercises, including both winning and losing stimuli, to facilitate their understanding of the experimental procedure. A complete round of the task would last 18 min 40 s.

### 2.3. Data Acquisition

The acquisition system consisted of two parts: the EEG data recording room and the MRI data scanning room. The synchronized data acquired included EEG, ECG, fMRI, and sMRI. A clock synchronization box was used to ensure the synchronicity of the data. EEG data were recorded by Net-Station (Electrical Geodesics Inc., Eugene, OR, USA) according to the international 10–10 electrode distribution system using a 64-channel MR-compatible electrode cap (HydroCel Geodesic Sensor Net; Electrical Geodesics, Inc., Eugene, OR, USA), sampled at 250 Hz, with Cz reference. In the data collection period, the impedance of all electrodes was kept below 50 kΩ. MRI data were acquired on 3T MR scanners (Philips Medical Systems, Best, The Netherlands) with an echo time (TE) of 35 ms, a repetition time (TR) of 2 s, and a flip angle of 90°. Twenty-four consecutive slices were scanned sequentially with a slice thickness of 4 mm and a field of view (FOV) of 230 × 180 mm^2^. The voxel size of the structural MRI images was 1 × 1 × 1 mm^3^.

The experiments were conducted in the hospital’s MRI scanning room. Electrode caps were MR-compatible, and the amplifier was placed in the MRI scanning room. A clock synchronization box was used to ensure the synchronicity of the EEG and fMRI data. During the experiment, the subject lay flat in the MR scanner, watched the gambling task on the screen, and chose the door which he thought would win by pressing a button.

### 2.4. Data Preprocessing

The EEG data quality can be severely compromised when recording inside the MR environment [36], so denoising is necessary. The collected EEG data were preprocessed using EEGLAB [37] to remove artifacts such as fMRI gradient artifacts, pulse artifacts, and power-frequency interference. First, fMRI gradient and pulse artifacts were removed using the FMRIB plug-in for EEGLAB, provided by the University of Oxford Centre for Functional MRI of the Brain (FMRIB) [38,39]. Next, a bandpass filter with cutoff frequencies of 1 Hz and 30 Hz was applied to remove direct current drift and high-frequency artifacts unrelated to neuronal oscillations. A 50 Hz notch filter was applied to the data to remove electrical line noise. The fMRI scan markers (“TREV”) were used to segment the EEG data, and the baseline was corrected so that the number of EEG segments and fMRI volumes were equal. Then, the EEG data were average referenced. Finally, components related to blinking, motion, and components not associated with neurological data were removed by ICA [40].

The fMRI data were preprocessed with the DPABI toolbox [41]. First, the fMRI data were converted from the DICOM (Digital Imaging and Communications in Medicine) format to the NIfTI (Neuroimaging Informatics Technology Initiative) format. Then, the fMRI images were slice-time-corrected, motion-corrected, normalized to MNI (Montreal Neurologic Institute) space, and resampled to a voxel size of 3 × 3 × 3 mm^3^, smoothed using a Gaussian kernel of 8 mm full width at half maximum (FWHM), and filtered (0.01–0.08 Hz). Among them, fMRI data with head movement amplitude more than 2 mm horizontal movement or 2 degrees rotation angle were considered as poor data quality and were therefore discarded. Finally, 90 ROIs were extracted according to the AAL template [42], and the average activation intensity of each ROI was calculated separately to obtain one BOLD time series for each ROI.

### 2.5. Higher-Order Data Representation

To reveal the interdependence between different data dimensions, we represented the EEG and fMRI data as a third-order tensor and a two-dimensional matrix, respectively, and then decomposed them jointly. Due to its capacity to preserve the data’s inherent multiway properties, this method differs from the conventional joint decomposition.

A tensorization strategy based on time-frequency transformation was used to convert the EEG data into a third-order tensor (time points × frequencies × channels). A spectrogram for each segment (length equal to the repetition time of the fMRI data) of each channel was calculated as the third dimension of the EEG data using Thomson’s multitaper method [43]. From 1 Hz to 30 Hz, the squared Fourier amplitudes were averaged into 0.5 Hz bins. Thus, an EEG tensor $X\in R^{I_{s}\times I_{f}\times I_{c}}$ was obtained that was synchronized in time with the fMRI data. Figure 3 depicts the formation of the EEG tensor.

The fMRI data $Y\in R^{I_{s}\times I_{r}}$ were represented as a two-dimensional spatiotemporal matrix (time points × ROIs), storing the time series of each ROI. Since the EEG data were segmented according to the fMRI scan, i.e., the number of segments of the EEG data was the same as the number of volumes of the fMRI data, the EEG and fMRI data were aligned with each other in the “time” mode.

### 2.6. Modeling the Hemodynamic Response Function

Although EEG and fMRI data were acquired simultaneously, the electrophysiological changes corresponding to the same neural process captured by the EEG were much faster on the time scale compared to the sluggish BOLD signal fluctuations. The neurovascular coupling of the relationship between these two complementary signals can be described by convolution with an HRF. In this paper, the HRF was parametrically estimated from the data [44]: one of many methods for modeling the HRF. As shown in Figure 4, the contribution of the EEG source to the BOLD signal of the ROI was represented by convolving the EEG data with an a priori unknown, ROI-specific HRF.

### 2.7. Coupled Matrix Tensor Factorization

The CMTF method proposed by Van Eyndhoven et al. [29] jointly decomposes the EEG tensor X and the fMRI matrix Y into a set of sources (also called “components”). The CMTF model can be described as:(1)$$X=\&\hat{X}+\varepsilon_{x}=\&\overset{R}{\underset{r=1}{\sum }}s_{r}\circ f_{r}\circ c_{r}+\varepsilon_{x}=\&⟦S,F,C⟧+\varepsilon_{x}$$
(2)$$Y=\&\hat{Y}+E_{y}=\&\overset{R}{\underset{r=1}{\sum }}\overset{K}{\underset{k=1}{\sum }}\left(H_{k}s_{r}\right)\circ \left(b_{k}*v_{r}\right)+\overset{Q}{\underset{q=1}{\sum }}n_{q}\circ P_{q}+E_{y}=\&\overset{K}{\underset{k=1}{\sum }}\left(H_{k}S\right)\left(b_{k}^{T}⊙V^{T}\right)+NP^{T}+E_{y}=\&\left[H_{1}S\;\ldots \;H_{k}S\right]⊙\left[B^{T}⊙V^{T}\right]+⟦N,P⟧+E_{y}$$

$X\in R^{I_{s}\times I_{f}\times I_{c}}$ is the third-order EEG tensor (time points × frequencies × channels), where $I_{s}$ is the number of time points, $I_{f}$ is the number of frequency bins, and $I_{c}$ is the number of channels. $\hat{X}$ is the low-rank approximation of X, which is the sum of R rank-1 terms of the canonical polyadic decomposition (CPD) [45]. $\varepsilon_{x}$ is the residual. Each rank-1 term $s_{r}\circ f_{r}\circ c_{r}$ is the outer product (∘) of time points, frequencies, and channels’ signatures, denoting the component corresponding to a source. ⟦S,F,C⟧ describes the CPD model composed of factor matrices $S\in R^{I_{s}\times R}$, $F\in R^{I_{f}\times R}$, and $C\in R^{I_{c}\times R}$, which hold the temporal, spectral, and EEG spatial signatures in the columns.

$Y\in R^{I_{s}\times I_{v}}$ is the fMRI matrix (time points × ROIs), where $I_{s}$ is the number of time points and $I_{v}$ is the number of ROIs. $\hat{Y}$ is the low-rank approximation of Y and is the sum of R rank-1 terms. $E_{y}$ is the residual. The coupling arises from the temporal signature $s_{r}$ which is shared in the EEG and fMRI decompositions. $H_{k}$ is the HRF matrix, V is the fMRI spatial factor matrix, and B is the HRF basis coefficient matrix. The temporal signatures $s_{r}$ are weighted with the spatial signatures $v_{r}$ for each source’s BOLD temporal signatures after convolution by HRF. To accommodate the additional structural changes in the fMRI data unrelated to electrophysiological dynamics, a rank-Q low-rank term ⟦N,P⟧ that is not coupled to the EEG decomposition is added to the fMRI decomposition. The coupling component of Y is described using RK non-independent rank-1 terms. Each rank-1 term $\left(H_{k}s_{r}\right)\circ \left(b_{k}*v_{r}\right)$ represents the convolution of the temporal signatures of the rth source with the kth basis function.

The diagram in Figure 4 illustrates the decomposition method in Equations (1) and (2).
(3)$$J\left(S,F,C,B,V,\theta \right)=\&\beta_{x}\|X-\hat{X}{\|}_{F}^{2}+\beta_{y}\|Y-\hat{Y}{\|}_{F}^{2}+\gamma_{x}\|\lambda_{x}{\|}_{1}+\gamma_{y}\|\lambda_{y}{\|}_{1}$$
(4)$$\begin{array}{c} s.t.\;H_{k}=\&T\left(h_{k}\right)=T\left(H\left(\theta^{\left(k\right)}\right)\right)\\ \lambda_{x}=\&{\left[\lambda_{x,1}\;\ldots \;\lambda_{x,R}\right]}^{T}\\ \lambda_{x,r}=\&\|s_{r}{\|}_{2}\cdot \|f_{r}{\|}_{2}\cdot \|c_{r}{\|}_{2}\\ \lambda_{y}=\&{\left[\lambda_{y,1}\;\ldots \;\lambda_{y,R}\right]}^{T}\\ \lambda_{y,r}=\&\sum_{k=1}^{K}\|b_{k}*v_{r}{\|}_{2} \end{array}$$

Equation (3) is minimized iteratively to estimate all model parameters. The cost function J consists of two data fit terms and two regularization terms. $\beta_{x},\beta_{y},\gamma_{x},\gamma_{y}$ denote positive weights and $\lambda_{x},\lambda_{y}$ are vectors that maintain the amplitude of each source in the EEG and fMRI data, respectively. The squared Frobenius norm of the residuals gives a good fit of the low-rank estimation term to the data. The L1 regularization term penalizes excessive source amplitudes and refines the model.

### 2.8. Model Selection

Most algorithms have their theoretical global optimal solutions for adaptive BSS algorithms. The algorithms may, however, also converge to a locally optimal solution due to interference from issues such as noise. The CP decomposition of the tensor should ideally be the same for each decomposition, but in practice, the results are likely different. Since there is no guarantee that the cost function J can converge to the global optimum, it is crucial to choose a good starting point to obtain a reliable solution. To obtain an excellent initial factor ⟦S,F,C⟧ in the CMTF model, 50 times decompositions of the EEG data were performed according to the CP model using the Tensorlab toolbox [46] for different numbers of components R. The Gauss–Newton iteration method [47] (cpd_nls with 2000 iterations, 400 conjugate gradient iterations for the step computation, and tolerance on the relative cost function update of $10^{-8}$, in Tensorlab 3.0 (Vervliet N, Leuven., Belgium)) was run for each decomposition to find a stable solution using a random initial factor. The results of 50 independent CP decompositions were then used as the initialization for CMTF of the EEG and fMRI data to iteratively optimize Equation (3) using the Quasi–Newton methods [48] (sdf_minf with 1000 iterations, and tolerance on the relative cost function update of $10^{-8}$, in Tensorlab 3.0 (Vervliet N, Leuven., Belgium)). This method decomposed the EEG and fMRI data along a common temporal dimension. The data from both modalities shared temporal patterns, which were the coupling mechanisms in the decomposition.

The CMTF decomposed the joint EEG and fMRI data into R components, each with 4 factors, namely, a frequency factor corresponding to the EEG frequency dimension, a channel factor corresponding to the EEG channel dimension, a time factor corresponding to the EEG–fMRI coupling time dimension, and an ROI factor corresponding to the fMRI ROI dimension.

For each subject, the CMTF component set with the most appropriate number of components R needed to be selected.

As shown in Figure 5, for each number of components R, the results ⟦S,F,C,V⟧ of 50 runs of the CMTF were analyzed using the graph-structured clustering algorithm [49] to obtain multiple component clusters, with the higher cardinality of a cluster being the more stable and plausible. The central clustering component of the component cluster was used to represent that component cluster, and the component that best represented that component number R was selected.

The selection of component number R was performed using four indicators:CORCONDIA [50]: the CORCONDIA is computed for the EEG tensor in combination with EEG components ⟦S,F,C⟧, which describes how well the CP decomposition of a given number of components is appropriate for a given tensor and a given factor. Furthermore, 100% indicates an adequate model, and below 80% indicates an inappropriate model;Reproducibility of components: the cardinality of the component most relevant to the time course of the stimulus is calculated, and clusters of components with cardinality greater than 10 are retained;Similarity to the time course of the stimulus: the time course of the paradigm stimulus is constructed. If a stimulus (losing/winning) is available at a time point, it is 1. Otherwise, it is 0. The correlation between the time component and the time course of the paradigm stimulus is calculated, and the higher the correlation, the better the model fits;Significance of spatial components: a statistical non-parametric map (SnPM) is calculated based on the spatial signature $v_{r}$. The familywise error (FWE) rate is controlled by setting the significance threshold at α = 0.05. A higher statistical score indicates that the model is more suitable for the component.


It is worth noting that different models may score well on different indicators, so the ranking of the models is inevitably ambiguous.

## 3. Results

The best model was selected manually for each of the 20 subjects based on the four indicators of model selection upon different stimuli. The component in each model chosen as the stimulus-related component had the highest similarity to the time course of the stimulus. For the best model, statistical non-parametric maps (SnPMs) were created for the spatial signatures $v_{r}$ of the stimulus-related components to calculate ROI activation. For different stimuli, the occurrence of activated ROIs known to be associated with that stimulus was counted for all subjects. The activated ROIs were drawn using the BrainNet Viewer toolbox [51].

When stimulated by winning or losing, the sources associated with reward processing generated a neural signal. The frequency factor explains the frequency information of this neural signal, the channel and ROI factors explain the area of diffusion of this neural signal, and the temporal factor explains the intensity of this neural signal over time.

### 3.1. Winning and Losing Stimuli

The CMTF method was used to decompose the components associated with both winning and losing stimuli.

For subject 13, the four indicators of the CMTF model were calculated as shown in Figure 6. The *x*-value indicates the number of components used in the model. Based on the selection guidelines, a component number of two was chosen. Figure 7 shows the results of two components, the first of which was stimulus-related. The frequency spectrum of the stimulus-related component showed higher energy in both the theta band (4–8 Hz) and alpha band (8–12 Hz). The topographic and the activation maps of the brain regions were overall consistent. Activation in the olfactory cortex, insula, anterior cingulate and paracingulate gyrus, posterior cingulate gyrus, parahippocampal gyrus, amygdala, inferior occipital gyrus, and caudate nucleus could be seen. A small activation in the orbital middle frontal gyrus and the orbital inferior frontal gyrus was also observed.

The presences of activated ROIs known to be associated with both winning and losing stimuli were counted for all subjects, as shown in Table 2. It can be seen that activation was present in the orbitofrontal cortex of all subjects. Meanwhile, activation was present in the insula and amygdala in almost all subjects. In most subjects, activation was present in the anterior cingulate cortex and caudate nucleus. In addition, 18 subjects had activation in the olfactory cortex; 17 subjects had activation in the posterior cingulate gyrus, parahippocampal gyrus, inferior occipital gyrus, and fusiform gyrus; and 14 subjects had activation in the rectus gyrus.

### 3.2. Winning Stimulus

The CMTF method was used to decompose the components associated with the winning stimulus.

Figure 8 shows the calculated four indicators of the CMTF model for subject 16. A component number of two was selected following the model selection principles, of which the first was stimulus-related. The results are shown in Figure 9. The spectrum of the stimulus-related component had an energy peak in the delta band (0.5–4 Hz), whereas the second component peaked in the beta band (12–30 Hz). The topographic and ROI activation maps of the stimulus-related components were generally consistent. There was activation in the olfactory cortex, intraorbital superior frontal gyrus, rectus gyrus, insula, anterior cingulate and paracingulate gyrus, caudate nucleus, and putamen. There was a small amount of activation in the dorsolateral superior frontal gyrus, cuneus, amygdala, and supramarginal gyrus.

The occurrence of activated ROIs known to be associated with the winning stimulus was counted for all subjects, as shown in Table 3. Activation was present in almost all subjects’ anterior cingulate cortex and caudate nucleus. Activation was present in the supramarginal gyrus and the putamen in most subjects. A small number of subjects had activation in the DLPFC and cuneus. In addition, 18 subjects had activation in the amygdala; 17 in the rectus gyrus, olfactory cortex, intraorbital superior frontal gyrus, inferior occipital gyrus, and insula; and 16 subjects had activation in the fusiform gyrus and parahippocampal gyrus.

### 3.3. Losing Stimulus

The CMTF method was used to decompose the components relevant to the stimulation of loss.

The four indicators of the CMTF model of different numbers of components for subject 1 were calculated as shown in Figure 10. A component number of five was selected to analyze the CMTF results of this subject by the model selection principles. The results are shown in Figure 11. The second of the five components was the stimulus-related component. The frequency spectrum of the stimulus-related component showed an energy peak in the delta band (0.5–4 Hz). The first component showed an energy peak in the theta band (4–8 Hz). The rest of the components showed energy peaks in the beta band (12–30 Hz). The topographic maps of stimulus-related components and brain area activation maps were generally consistent. There was activation in the olfactory cortex, insula, posterior cingulate gyrus, parahippocampal gyrus, amygdala, and fusiform gyrus. There was a small activation in the triangle inferior frontal gyrus and orbital inferior frontal gyrus.

The occurrence of activated ROIs known to be associated with the losing stimulus was counted for all subjects, as shown in Table 4. Activation was present in the inferior frontal gyrus of all subjects. Activation was present in the cingulate cortex in almost all subjects. Activation was present in the insula of most subjects. In addition, there was activation in the rectus gyrus and amygdala in 18 subjects; the inferior occipital gyrus in 17 subjects; the parahippocampal gyrus in 16 subjects; and the fusiform gyrus, the olfactory cortex, and the medial superior frontal gyrus in 15 subjects.

## 4. Discussion

Reward processing was investigated by taking advantage of the CMTF method’s ability to capture the higher-order structure and the complementary nature of multimodal data. The results were presented in four dimensions: the temporal dimension, which reveals how reward is processed over time; the frequency dimension that shows the potential oscillatory rhythmicity of reward processing, which helps distinguish brain processes or functions by frequency bands; the channel dimension that examines the topography of scalp activity; and the ROI dimension which gives an idea about the regional activation.

The mesocorticolimbic pathway comprises the ventral striatum, nucleus accumbens, orbitofrontal cortex, dorsal striatum (including the caudate nucleus, etc.), medial prefrontal cortex, and amygdala. It has been revealed that it is heavily involved in reward processing [52]. Using the CMTF analysis for both winning and losing stimuli, activation was found in the orbitofrontal cortex for all subjects; in the insula, olfactory cortex, posterior cingulate gyrus, parahippocampal gyrus, inferior occipital gyrus, fusiform gyrus, and amygdala for almost all subjects; and in the rectus gyrus, anterior cingulate cortex, and caudate nucleus for most subjects. While the result agrees well with the literature [53,54,55], it does unveil some information that has never been reported before. For example, we found that the olfactory cortex, inferior occipital gyrus, and fusiform gyrus were also activated when winning and losing, which was not mentioned in other studies. The olfactory cortex is part of the limbic system, which is involved in processing emotions, forming memories, and linking the senses to memories and emotions [56]. This could be the reason why it was activated. In the meantime, the inferior occipital and fusiform gyrus were activated most likely because they are associated with vision, such as processing color information, face and body recognition, and word recognition [57,58], which were all heavily involved in the gambling task utilized in this study.

When comparing winning and losing stimuli, we found that the former led to greater brain activation in parts of the reward system and areas involved in reward anticipation, decision-making, and impulse control. For example, activation was present in the superior orbital frontal gyrus and part of the reward system (anterior cingulate cortex and caudate nucleus) for almost all subjects; in the supramarginal gyrus and the putamen for most subjects; and in the cuneus for a small number of subjects. In contrast, activation in risk aversion and uncertainty management areas was increased for the losing stimulus. For instance, activation was present in the inferior frontal gyrus for all subjects, the cingulate cortex for almost all subjects, and the medial superior frontal gyrus for most subjects. The result agrees well with the one from McClure et al. [52], where activation was spotted in the caudate nucleus and anterior cingulate cortex when winning but in the inferior frontal gyrus when losing. Moreover, Knutson et al. [59] found that expected wins led to increased activation in the cuneus and caudate nucleus, but unexpected losses activated the insula. Yazdi et al. [60] found that unexpected wins were associated with increased brain activation in the supramarginal gyrus and the putamen, which are regions associated with surprise. These findings are all consistent with the results of this paper.

In comparison with the EEG-informed fMRI analysis of the fusion results [23], the CMTF method showed stronger activation for both winning and losing stimuli in both insula and olfactory cortex. For winning stimuli, greater activation was shown in the supramarginal gyrus. In addition, activation of the inferior occipital gyrus and fusiform gyrus during winning and losing was found, possibly because they are associated with vision.

Overall, the CMTF successfully extracted meaningful components associated with stimuli. The CMTF can estimate features and statistical maps of multiple components, which provides a powerful advantage over a classical EEG/fMRI correlation analysis. As we demonstrated in our experiments, artificial influences may be isolated in separate components, which can reduce their impact on stimulus mapping in the brain.

## 5. Conclusions

This paper investigates the simultaneous EEG–fMRI data of reward processing using the CMTF method. The CMTF method takes advantage of multimodal data complementarity and preservation of interdependencies of different dimensions to simultaneously study the neural activity of the human brain during reward processing from three perspectives: temporal, frequency, and spatial. Not only were the activations in ROIs found in other literature discovered, but also in the olfactory cortex and fusiform gyrus, which were previously ignored. The results help explore the neurovascular coupling of reward processing, and also improve the understanding of brain function. Note that this study assumes that the factors shared by the EEG third-order tensor and the fMRI matrix are identical. This might not be the exact case, even though they do share similar covariances. The sample size is also relatively small due to the complexity and difficulty of EEG–fMRI fusion experiments. A new objective function to capture the shared components in EEG and fMRI will be studied and constructed in future works.

## Figures and Tables

**Figure 1 brainsci-13-00485-f001:**
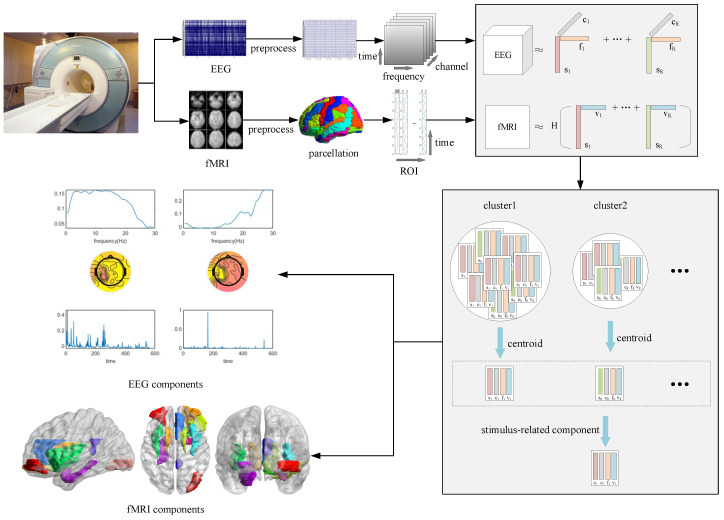
Using the CMTF method, the simultaneous EEG-fMRI data were jointly decomposed to capture ROIs activated by reward processing. The collected data went through preprocessing, dimensional expansion of EEG data, CMTF, and model selection. After using graph-structured clustering, the best model was selected by four quantitative metrics, CORCONDIA, reproducibility, similarity, and significance, to obtain the fMRI components (ROI factors) and the corresponding EEG components (frequency factors, channel factors, and time factors).

**Figure 2 brainsci-13-00485-f002:**
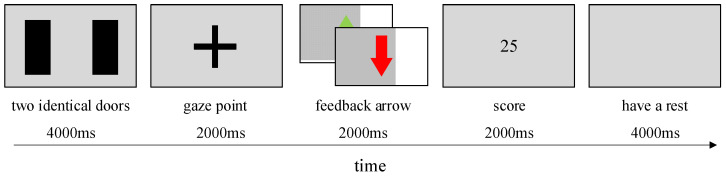
Single-trial flow is as follows. Two identical doors would appear simultaneously on the computer screen, one corresponding to a win (+USD 2.0) and the other to a loss (−USD 1.0). Subjects were instructed beforehand to always choose the door they thought would win money using an MRI-compatible response box. The computer would randomly select a door if the subject did not make a timely choice. A 2000 ms gaze point and a 2000 ms feedback arrow came afterward. A green up arrow indicated a win, and a red down arrow a loss. At the end of the arrow feedback, the center of the screen would display the subject’s current cumulative score. After each trial, subjects were given a 4000 ms break.

**Figure 3 brainsci-13-00485-f003:**
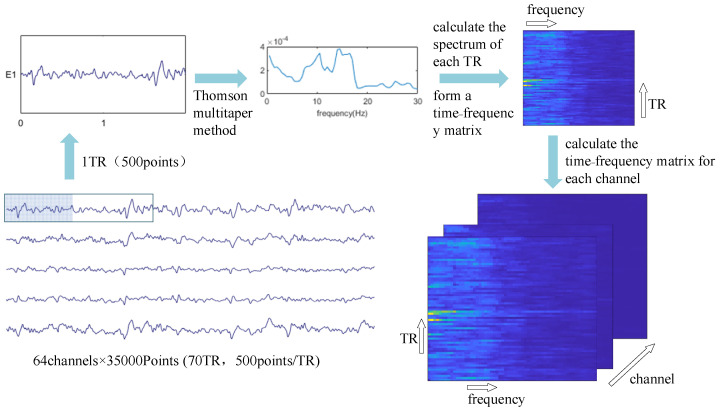
A tensorization strategy based on time-frequency transformation was used to convert the EEG data into a third-order tensor (time points × frequencies × channels). A spectrogram for each segment (length equal to the repetition time of the fMRI data) of each channel was calculated as the third dimension of the EEG data using Thomson’s multitaper method. From 1 Hz to 30 Hz, the squared Fourier amplitudes were averaged into 0.5 Hz bins. Thus, an EEG tensor was obtained and was synchronized in time with the fMRI data.

**Figure 4 brainsci-13-00485-f004:**
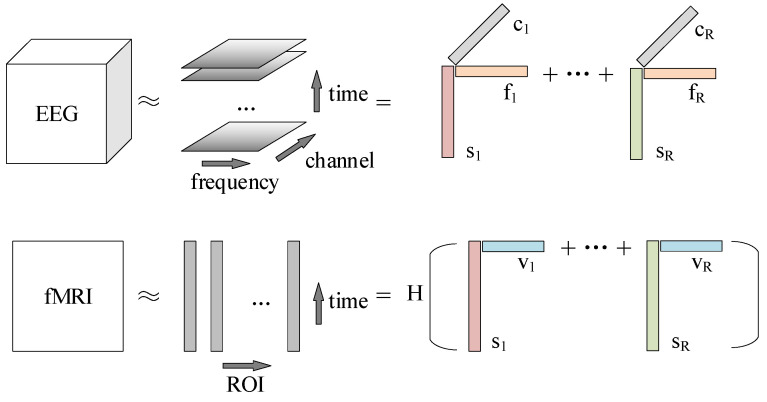
The CMTF method decomposes the sources shared in both modal data. The EEG tensor is decomposed into R components, each consisting of a temporal signature $s_{r}$, spectral signature $f_{r}$, and spatial (channel) signature $c_{r}$. The fMRI data are decomposed into a convolution of R components and HRF, each consisting of a temporal signature $s_{r}$ and spatial (ROI) signature $v_{r}$.

**Figure 5 brainsci-13-00485-f005:**
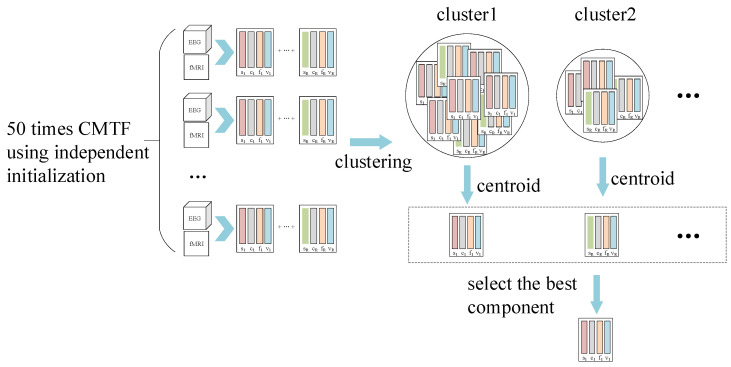
For each number of components R, the results ⟦S,F,C,V⟧ of 50 runs of the CMTF were analyzed using the graph-structured clustering algorithm to obtain multiple component clusters, with the higher cardinality of a cluster being the more stable and plausible. The central clustering component of the component cluster was used to represent that component cluster, and the component that best represented that component number R was selected.

**Figure 6 brainsci-13-00485-f006:**
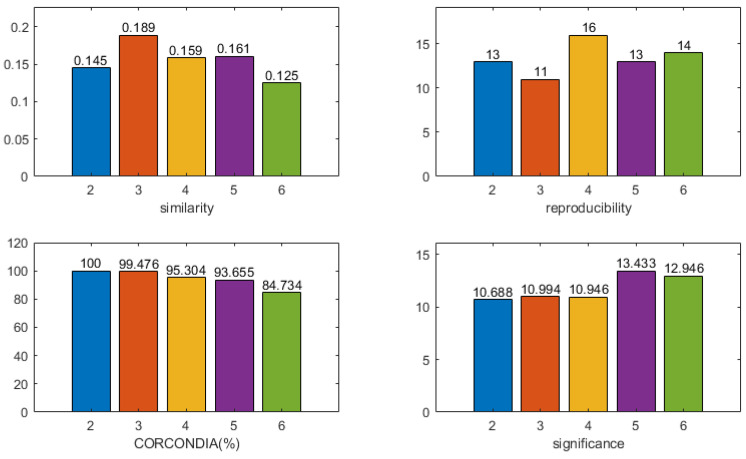
The four indicators of the CMTF model of different numbers of components for subject 13 were calculated. According to the model selection guidelines, a component number of 2 was chosen to analyze the CMTF results for this subject.

**Figure 7 brainsci-13-00485-f007:**
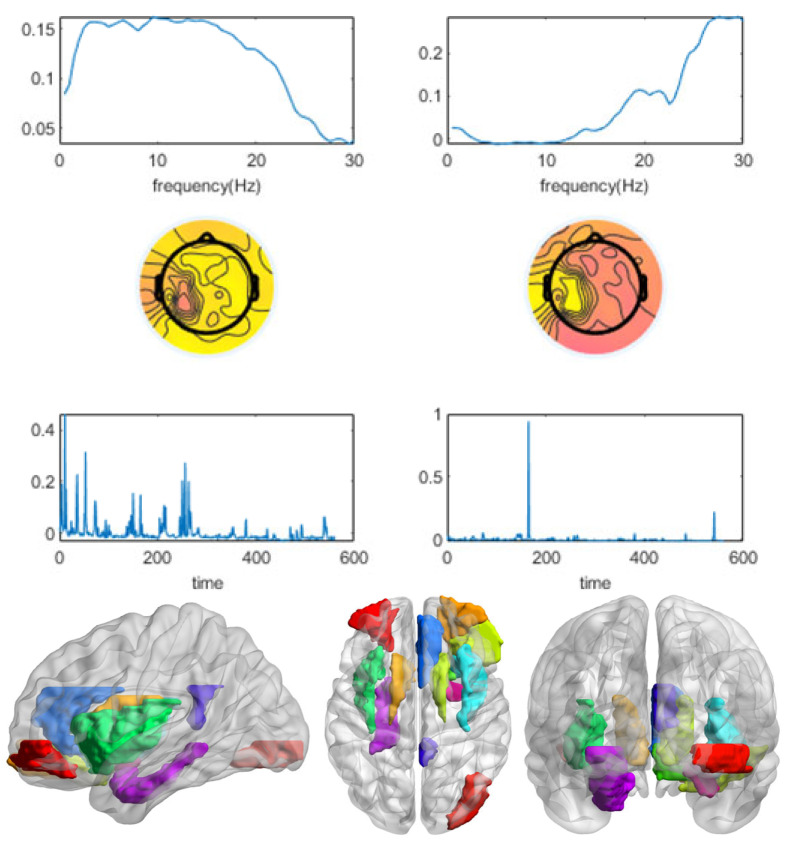
The CMTF results for subject 13 demonstrate the temporal signature $s_{r}$, spectral signature $f_{r}$, and spatial (channel) signature $c_{r}$ of 2 sources in the EEG domain and activation of the ROI associated with both winning and losing stimuli.

**Figure 8 brainsci-13-00485-f008:**
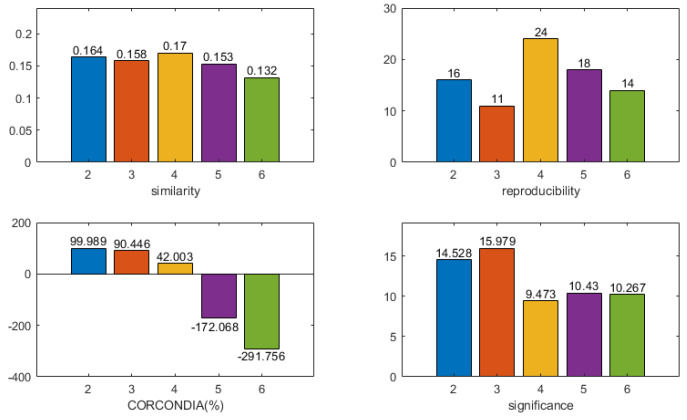
The four indicators of the CMTF model of different numbers of components for subject 16 were calculated. According to the model selection guidelines, a component number of 2 was chosen to analyze the CMTF results for this subject.

**Figure 9 brainsci-13-00485-f009:**
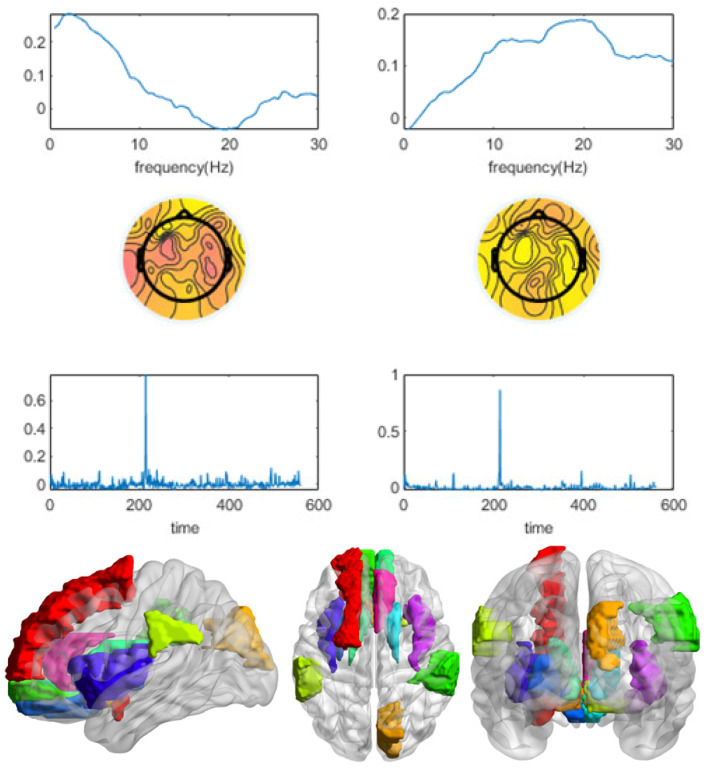
The CMTF results for subject 16 demonstrate the temporal signature $s_{r}$, spectral signature $f_{r}$, and spatial (channel) signature $c_{r}$ of 2 sources in the EEG domain and ROI activation associated with winning stimuli.

**Figure 10 brainsci-13-00485-f010:**
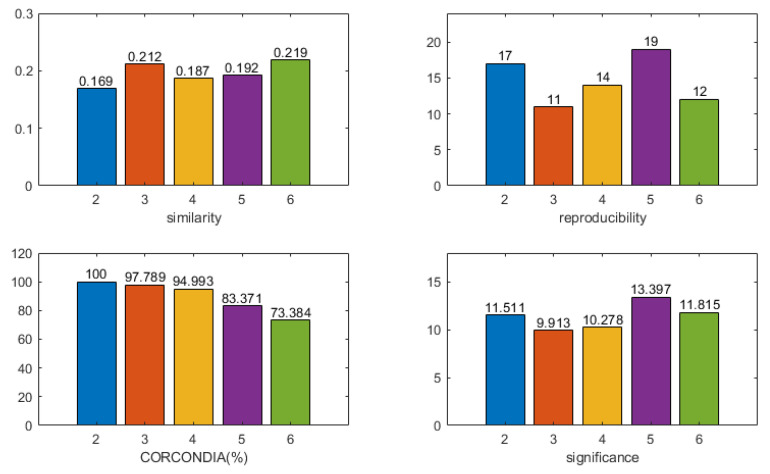
The four indicators of the CMTF model of different numbers of components for subject 1 were calculated. According to the model selection guidelines, a component number of 5 was chosen to analyze the CMTF results for this subject.

**Figure 11 brainsci-13-00485-f011:**
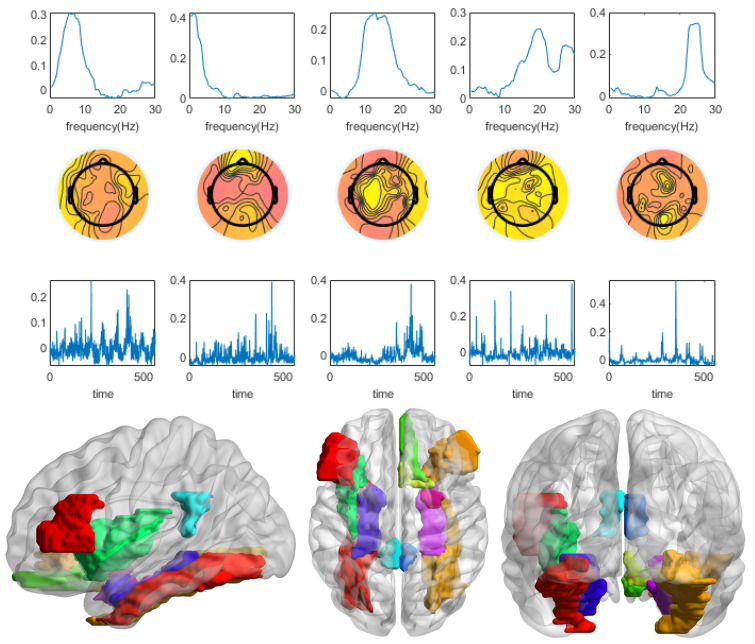
The CMTF results for subject 1 demonstrate the temporal signature $s_{r}$, spectral signature $f_{r}$, and spatial (channel) signature $c_{r}$ of 5 sources in the EEG domain and ROI activation associated with the losing stimulus.

**Table 1 brainsci-13-00485-t001:** Subject data.

Patient	Gender	Age	Degree of Myopia	Dominant Hand
Sub 01	male	21	no myopia	right
Sub 02	male	23	no myopia	right
Sub 03	female	19	300 (left, right)	right
Sub 04	male	20	no myopia	right
Sub 05	male	22	no myopia	right
Sub 06	male	21	no myopia	right
Sub 07	male	21	no myopia	right
Sub 08	male	22	no myopia	right
Sub 09	male	23	no myopia	right
Sub 10	male	22	no myopia	right
Sub 11	male	22	no myopia	right
Sub 12	male	20	300 (left, right)	right
Sub 13	female	20	no myopia	right
Sub 14	male	21	275 (left, right)	right
Sub 15	male	25	no myopia	right
Sub 16	male	23	no myopia	right
Sub 17	male	21	250 (right), 150 (right)	right
Sub 18	male	25	no myopia	right
Sub 19	female	22	no myopia	right
Sub 20	male	22	no myopia	right

**Table 2 brainsci-13-00485-t002:** Partial activated ROIs of all subjects of both winning and losing stimuli.

Subject	Orbitofrontal Cortex	Insula	Anterior Cingulate Cortex	Amygdala	Caudate Nucleus
Sub 01	√	√	√	√	
Sub 02	√	√	√	√	√
Sub 03	√	√	√	√	
Sub 04	√	√		√	√
Sub 05	√	√	√	√	
Sub 06	√		√	√	√
Sub 07	√	√		√	√
Sub 08	√	√	√		√
Sub 09	√	√	√	√	
Sub 10	√	√		√	√
Sub 11	√				√
Sub 12	√	√		√	√
Sub 13	√	√	√	√	√
Sub 14	√	√	√	√	
Sub 15	√	√		√	√
Sub 16	√	√	√	√	√
Sub 17	√	√	√	√	√
Sub 18	√	√	√	√	√
Sub 19	√			√	√
Sub 20	√	√			

**Table 3 brainsci-13-00485-t003:** Partial activated ROIs of all subjects of winning stimulus.

Subject	DLPFC	Anterior Cingulate Cortex	Cuneus	Supramarginal Gyrus	Caudate Nucleus	Putamen
Sub 01		√	√			√
Sub 02		√			√	
Sub 03		√	√		√	
Sub 04		√			√	
Sub 05	√	√		√	√	
Sub 06		√			√	
Sub 07		√		√	√	
Sub 08		√	√	√	√	√
Sub 09	√	√		√	√	√
Sub 10		√		√	√	√
Sub 11	√	√		√	√	√
Sub 12	√	√		√	√	√
Sub 13		√	√		√	√
Sub 14	√	√	√			
Sub 15	√	√			√	√
Sub 16	√	√	√	√	√	√
Sub 17		√		√	√	
Sub 18	√	√	√	√	√	
Sub 19	√	√		√	√	√
Sub 20				√		√

**Table 4 brainsci-13-00485-t004:** Partial activated ROIs of all subjects of losing stimulus.

Subject	Inferior Frontal Gyrus	Insula	Cingulate Cortex
Sub 01	√	√	√
Sub 02	√	√	√
Sub 03	√	√	√
Sub 04	√	√	√
Sub 05	√	√	√
Sub 06	√		√
Sub 07	√		√
Sub 08	√	√	√
Sub 09	√	√	√
Sub 10	√	√	√
Sub 11	√	√	√
Sub 12	√		√
Sub 13	√	√	√
Sub 14	√	√	√
Sub 15	√	√	√
Sub 16	√	√	
Sub 17	√	√	
Sub 18	√	√	√
Sub 19	√	√	√
Sub 20	√		√

## Data Availability

Data are unavailable due to privacy restrictions.

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
