# Peer review of "Exploring Neural Mechanisms of Reward Processing Using Coupled Matrix Tensor Factorization: A Simultaneous EEG–fMRI Investigation"

_brainsci, 2023, doi:10.3390/brainsci13030485_

Round 1

Reviewer 1 Report

This is an interesting paper on neural mechanisms of reward processing using coupled matrix tensor factorization. The topic is interesting and I agree that it may have a good contribution to the literature. However, I have several comments to improve the manuscript further:

1. First, the paper will be benefited from more proofreading and rewriting as many of the sentences are read awkwardly. For example, in the abstract: ' It is crucial for reward processing to understand the brain's neural feedback mechanisms and cognitive decision-making." Similarly, this sentence very awkwardly and unclear: "In this paper, the unknown neurovascular coupling between electrophysiological phenomena measured by  EEG  and hemodynamic changes captured by  blood  oxygen level-dependent (BOLD) signals is explained using the hemodynamic response function (HRF)." They are difficult to read. I would like to encourage the authors to ensure that the paper is readable before it can be published. A major revision is necessary.

2. It will be useful for the authors to supplement more information regarding the participants' characteristics beyond just gender, age, and handedness.

3. The inclusion and exclusion criteria of the participants should be reported clearly. 

4. There should be more discussion on how coupled matrix tensor factorization is comparable to other more well-established or commonly used methodology

5. The sample size of the current study is relatively small and this should be acknowledged and highlighted in the discussion.

Author Response

Responses to reviewers’ comments of manuscript ID brainsci-2267779

Reviewer: 1

Comments to the Author

Point: This is an interesting paper on neural mechanisms of reward processing using coupled matrix tensor factorization. The topic is interesting and I agree that it may have a good contribution to the literature. However, I have several comments to improve the manuscript further:

Response: We thank the referee for reviewing our manuscript and appreciate her/his recognition of our work.

Point 1: First, the paper will be benefited from more proofreading and rewriting as many of the sentences are read awkwardly. For example, in the abstract: ' It is crucial for reward processing to understand the brain's neural feedback mechanisms and cognitive decision-making." Similarly, this sentence very awkwardly and unclear: "In this paper, the unknown neurovascular coupling between electrophysiological phenomena measured by EEG and hemodynamic changes captured by blood oxygen level-dependent (BOLD) signals is explained using the hemodynamic response function (HRF)." They are difficult to read. I would like to encourage the authors to ensure that the paper is readable before it can be published. A major revision is necessary.

Response 1: We have re-written the sentences mentioned by the referee as follows:

  • We have changed “It is crucial for reward processing to understand the brain's neural feedback mechanisms and cognitive decision-making.” to “It is crucial to understand the neural feedback mechanisms and the cognitive decision-making of the brain during the processing of rewards.” (In abstract, lines 13-15)
  • We have changed “In this paper, the unknown neurovascular coupling between electrophysiological phenomena measured by EEG and hemodynamic changes captured by blood oxygen level-dependent (BOLD) signals is explained using the hemodynamic response function (HRF).” to “In this paper, the unknown neurovascular coupling between electrophysiological phenomena, measured by EEG, and hemodynamic changes, captured by blood oxygen level-dependent (BOLD) signals, is explained using the hemodynamic response function (HRF).” (In section 1, paragraph 8, lines 138-140)

In addition, we have also heavy edited the wording and flow throughout the text. We hope by doing so would relieve the referee’s concern.

Point 2: It will be useful for the authors to supplement more information regarding the participants' characteristics beyond just gender, age, and handedness.

Response 2: The reward processing experiments in this study only involves healthy subjects, so no more medical indicators were recorded. This is also the case for many related articles on reward processing, where just gender, age, and handedness were provided (Li et al. 2010; Guo et al. 2017; Dong et al. 2018; Yazdi et al. 2019). It would be helpful if the referee can specify what other characteristics should be suppled. We note there are indeed other EEG-fMRI fusion studies provided more detailed information on subjects. For example,

  • Van Eyndhoven et al.(Van Eyndhoven et al. 2021) provided clinical patient data, such as gender, ictal onset zone, etiology and surgery.
  • Qin et al.(Qin et al. 2019) provided detailed demographic information and clinical characteristics of JME patients, such as gender, age, frequency of GSWDs, age at seizure onset, family history, AEDs and seizure frequency.

However, the subjects in these studies were all patients with a particular disease, which is not the case here in our work.

Point 3: The inclusion and exclusion criteria of the participants should be reported clearly.

Response 3: We have taken the referee’s advice by adding the inclusion and exclusion criteria of the participants in subsection 2.1, paragraph 1, lines 171-177 according to (Poldrack et al. 2008) as follows:

  • “The following inclusion criteria apply: (1) 18 to 25 years of age; (2) normal or corrected normal vision; (3) right-handed.”
  • “The following exclusion criteria apply: (1) use of corticosteroids and psychotic drugs within the past 30 days; (2) current or previous history of neurological, medical, or psychiatric disorders; (3) current or previous history of neurosurgery, head injury, cerebrovascular injury, or traumatic brain injury involving loss of consciousness; (4) presence of learning disability; (5) presence of claustrophobia; (6) refusal to give informed consent; and (7) presence of magnetic implants in the body.”

Point 4: There should be more discussion on how coupled matrix tensor factorization is comparable to other more well-established or commonly used methodology.

Response 4: We thank the referee for the suggestion. To satisfy the referee, we have added the disadvantages of other methods, the advantages of tensor factorization and a comparison of CMTF method with another method(Guo et al. 2017) in terms of reward processing results as follows:

  • “BSS methods that use matrices as input, such as ICA, can separate multivariate signals into additive components based on the assumption that they are statistically independent. However, this approach is deficient in exploiting the inherent multiway nature of these data(Hunyadi et al. 2017). Studies in only two dimensions, temporal and spatial, cannot reveal the inherent higher-order structure of the data nor the interdependence of the different dimensions. EEG and fMRI data are inherently multidimensional, which means it includes information about time, different voxels or channels, subjects, trials, etc. For EEG, the signal can be expanded along additional patterns in order to reveal more potential information. This multidimensional nature of EEG and fMRI data suggests the use of tensor models rather than matrix models.” (In section 1, paragraph 4, lines 97-106)
  • “Tensor decomposition models can improve the ability to extract spatiotemporal patterns of interest, facilitate neurophysiologically meaningful interpretations and produce unique representations under mild conditions(Chatzichristos et al. 2022). In addition, tensor methods are able to make predictions more robustly in the presence of noise than matrix-based methods(Chatzichristos et al. 2019).” (In section 1, paragraph 5, lines 109-112)
  • “In comparison with EEG-informed fMRI analysis of fusion results(Guo et al. 2017), the CMTF method showed stronger activation for both winning and losing stimuli in both insula and olfactory cortex. For winning stimuli, greater activation was shown in the supramarginal gyrus. In addition, activation of the inferior occipital gyrus and fusiform gyrus during winning and losing was found, possibly because they are associated with vision.” (In section 4, paragraph 4, lines 556-560)

Point 5: The sample size of the current study is relatively small and this should be acknowledged and highlighted in the discussion.

Response 5: We agree with the referee that the current sample size is relatively small. This is due to the complexity and difficulty of EEG-fMRI fusion experiments and the need for multiple departments to collaborate. In fact, we have already spent over a year to collect what we currently have. We have added the corresponding statement, and hope by doing so would relieve the referee’s concern:

“The sample size is also relatively small due to the complexity and difficulty of EEG-fMRI fusion experiments.” (In section 5, paragraph 1, lines 579-580)

References

Chatzichristos C, Kofidis E, Van Paesschen W, De Lathauwer L, Theodoridis S, Van Huffel S. Early soft and flexible fusion of electroencephalography and functional magnetic resonance imaging via double coupled matrix tensor factorization for multisubject group analysis. Human Brain Mapping. 2022 Mar;43(4):1231–55.

Chatzichristos C, Vandecapelle M, Kofidis E, Theodoridis S, Lathauwer LD, Van Huffel S. Tensor-based Blind fMRI Source Separation Without the Gaussian Noise Assumption — A β-Divergence Approach. In: 2019 IEEE Global Conference on Signal and Information Processing (GlobalSIP). 2019. p. 1–5.

Dong G, Li H, Wang Y, Potenza MN. Individual differences in self-reported reward-approach tendencies relate to resting-state and reward-task-based fMRI measures. International Journal of Psychophysiology. 2018 Jun 1;128:31–9.

Guo Q, Zhou T, Li W, Dong L, Wang S, Zou L. Single-trial EEG-informed fMRI analysis of emotional decision problems in hot executive function. Brain and Behavior. 2017;7(7):e00728.

Haas B de, Sereno MI, Schwarzkopf DS. Inferior Occipital Gyrus Is Organized along Common Gradients of Spatial and Face-Part Selectivity. J Neurosci. 2021 Jun 23;41(25):5511–21.

Hunyadi B, Dupont P, Van Paesschen W, Van Huffel S. Tensor decompositions and data fusion in epileptic electroencephalography and functional magnetic resonance imaging data: Tensors in EEG-fMRI. WIREs Data Mining Knowl Discov. 2017 Jan;7(1):e1197.

Li X, Lu ZL, D’Argembeau A, Ng M, Bechara A. The Iowa Gambling Task in fMRI images. Human Brain Mapping. 2010;31(3):410–23.

McDonald AJ. Amygdala. In: Aminoff MJ, Daroff RB, editors. Encyclopedia of the Neurological Sciences (Second Edition) [Internet]. Oxford: Academic Press; 2014 [cited 2022 Sep 29]. p. 153–6. Available from: https://www.sciencedirect.com/science/article/pii/B9780123851574011131

Mosayebi R, Hossein-Zadeh GA. Correlated coupled matrix tensor factorization method for simultaneous EEG-fMRI data fusion. Biomedical Signal Processing and Control. 2020 Sep;62:102071.

Poldrack RA, Fletcher PC, Henson RN, Worsley KJ, Brett M, Nichols TE. Guidelines for reporting an fMRI study. NeuroImage. 2008 Apr 1;40(2):409–14.

Qin Y, Jiang S, Zhang Q, Dong L, Jia X, He H, et al. Ballistocardiogram artifact removal in simultaneous EEG-fMRI using generative adversarial network. NeuroImage: Clinical. 2019;22:101759.

Van Eyndhoven S, Dupont P, Tousseyn S, Vervliet N, Van Paesschen W, Van Huffel S, et al. Augmenting interictal mapping with neurovascular coupling biomarkers by structured factorization of epileptic EEG and fMRI data. NeuroImage. 2021 Mar;228:117652.

Weiner KS, Zilles K. The anatomical and functional specialization of the fusiform gyrus. Neuropsychologia. 2016 Mar 1;83:48–62.

Yazdi K, Rumetshofer T, Gnauer M, Csillag D, Rosenleitner J, Kleiser R. Neurobiological processes during the Cambridge gambling task. Behavioural Brain Research. 2019 Jan 1;356:295–304.

Reviewer 2 Report

In order to understand the neurofeedback mechanism of the brain, this article is the first attempt to apply tensor decomposition to synchronize brain waves and fMRI in simulated gambling tasks. This is an interesting research paper. There are some suggestions for revision.

1.     The motivation is not clear. Please specify the importance of the proposed solution.

2.     Please highlight the contributions of this paper in introduction.

3.     The related work is a little bit weak. The authors ignore some solutions published in 2022 and 2023. For example, “Brain tumor segmentation based on the fusion of deep semantics and edge information in multimodal MRI” Information Fusion, Volume 91, 376-387, 2023 and "Dynamic functional connectivity estimation for neurofeedback emotion regulation paradigm with simultaneous EEG-fMRI analysis", Front. Hum. Neurosci., 16 September 2022, Sec. Cognitive Neuroscience, Volume 16 - 2022, https://doi.org/10.3389/fnhum.2022.933538.

4.     The introduction part of the article introduces some existing methods, but does not explain the problems of these methods, so the reasons for introducing the method in this paper are not sufficient. It is recommended to cite the existing problems to elicit the proposed method.

5.     The article lacks verification of why tensor decomposition can be used or better for reward mechanisms. The advantages of using tensor decomposition should be explained in the introduction

6.     The ROI in the article may be misunderstood by others, so the full name should be marked first. At the same time, can the transformation of fMRI be represented as in Figure 3?

7.     Symbols in some formulas in the text are not fully explained. For example, S, F, C in Formula 1; H, P in Formula 2; beta, gamma, etc. in Formula 3. Formulas should be checked and commented.

8.     Why is the Gauss-Newton iterative method used in model selection and what are the advantages of using it?

9.     Please discuss how to obtain the suitable parameter values used in the proposed solution.

10.  What is the experimental environment?

11.  The experimental results are not convincing. Please the proposed solution with more recently published solutions.

Author Response

Responses to reviewers’ comments of manuscript ID brainsci-2267779

Reviewer: 2

Comments to the Author

Point: In order to understand the neurofeedback mechanism of the brain, this article is the first attempt to apply tensor decomposition to synchronize brain waves and fMRI in simulated gambling tasks. This is an interesting research paper. There are some suggestions for revision.

Response: We thank the referee for reviewing our manuscript and recognizing the value of our work.

Point 1: The motivation is not clear. Please specify the importance of the proposed solution.

Response 1: Matrix factorization methods employ only two dimensions and cannot extract the interdependency between different dimensions of data. On the other side, tensor analysis approaches have the ability to provide a natural representation of multidimensional neuroimaging data(Mosayebi and Hossein-Zadeh 2020). So, it is necessary to use tensor models rather than matrix models. We have added the importance of the proposed solution as follows:

  • “BSS methods that use matrices as input, such as ICA, can separate multivariate signals into additive components based on the assumption that they are statistically independent. However, this approach is deficient in exploiting the inherent multiway nature of these data(Hunyadi et al. 2017). Studies in only two dimensions, temporal and spatial, cannot reveal the inherent higher-order structure of the data nor the interdependence of the different dimensions. EEG and fMRI data are inherently multidimensional, which means it includes information about time, different voxels or channels, subjects, trials, etc. For EEG, the signal can be expanded along additional patterns in order to reveal more potential information. This multidimensional nature of EEG and fMRI data suggests the use of tensor models rather than matrix models.” (In section 1, paragraph 4, lines 97-106)
  • “Tensor decomposition models can improve the ability to extract spatiotemporal patterns of interest, facilitate neurophysiologically meaningful interpretations and produce unique representations under mild conditions(Chatzichristos et al. 2022). In addition, tensor methods are able to make predictions more robustly in the presence of noise than matrix-based methods(Chatzichristos et al. 2019).” (In section 1, paragraph 5, lines 109-112)

Point 2: Please highlight the contributions of this paper in introduction.

Response 2: We have added the contributions of this paper in section 1, paragraph 8, lines 149-153 as follows:

“It is the first attempt to apply tensor decomposition to simultaneous EEG and fMRI data in a gambling task. The fact that CMTF method can reveal the inherent higher-order structure of the data and the interdependence of the different dimensions, makes it possible to discover additional ROIs which were previously ignored in other literature during the processing of rewards.”

Point 3: The related work is a little bit weak. The authors ignore some solutions published in 2022 and 2023. For example, “Brain tumor segmentation based on the fusion of deep semantics and edge information in multimodal MRI” Information Fusion, Volume 91, 376-387, 2023 and "Dynamic functional connectivity estimation for neurofeedback emotion regulation paradigm with simultaneous EEG-fMRI analysis", Front. Hum. Neurosci., 16 September 2022, Sec. Cognitive Neuroscience, Volume 16 - 2022, https://doi.org/10.3389/fnhum.2022.933538.

Response 3: We thank the referee for providing the references. We now have added them to the manuscript.

  • “Mosayebi et al.[33] employed the CCMTF method for analysis of the emotion regulation paradigm.” was added in section 1, paragraph 6, lines 130-131.
  • “…, such as the emerging field of deep learning-based fusion[14].” was added in section 1, paragraph 3, lines 82-83.

Point 4: The introduction part of the article introduces some existing methods, but does not explain the problems of these methods, so the reasons for introducing the method in this paper are not sufficient. It is recommended to cite the existing problems to elicit the proposed method.

Response 4: We have taken the referee’s advice by citing the existing problems of these methods in section 1, paragraph 4, lines 97-106 as follows, and hope it would relieve the referee’s concern:

“BSS methods that use matrices as input, such as ICA, can separate multivariate signals into additive components based on the assumption that they are statistically independent. However, this approach is deficient in exploiting the inherent multiway nature of these data(Hunyadi et al. 2017). Studies in only two dimensions, temporal and spatial, cannot reveal the inherent higher-order structure of the data nor the interdependence of the different dimensions. EEG and fMRI data are inherently multidimensional, which means it includes information about time, different voxels or channels, subjects, trials, etc. For EEG, the signal can be expanded along additional patterns in order to reveal more potential information. This multidimensional nature of EEG and fMRI data suggests the use of tensor models rather than matrix models.”

Point 5: The article lacks verification of why tensor decomposition can be used or better for reward mechanisms. The advantages of using tensor decomposition should be explained in the introduction.

Response 5: We have taken the referee’s advice by adding the advantages in section 1, paragraph 5, lines 109-112 as follows:

“Tensor decomposition models can improve the ability to extract spatiotemporal patterns of interest, facilitate neurophysiologically meaningful interpretations and produce unique representations under mild conditions(Chatzichristos et al. 2022). In addition, tensor methods are able to make predictions more robustly in the presence of noise than matrix-based methods(Chatzichristos et al. 2019).”

Point 6: The ROI in the article may be misunderstood by others, so the full name should be marked first. At the same time, can the transformation of fMRI be represented as in Figure 3?

Point 6-1: The ROI in the article may be misunderstood by others, so the full name should be marked first.

Response 6-1: We have added the full name of ROI in line 24 and 148.

Point 6-2: At the same time, can the transformation of fMRI be represented as in Figure 3?

Response 6-2: Figure 3 is a schematic representation of the expansion of the EEG into an EEG tensor rather than the fMRI. To avoid future misunderstanding, we have moved Figure 3 to the corresponding position in subsection 2.5, lines 271-277.

Point 7:  Symbols in some formulas in the text are not fully explained. For example, S, F, C in Formula 1; H, P in Formula 2; beta, gamma, etc. in Formula 3. Formulas should be checked and commented.

Response 7: We have gone throughout the text and supplied the necessary information. (In subsection 2.7, lines 307-311, lines 315-323, and lines 332-336)

Point 8: Why is the Gauss-Newton iterative method used in model selection and what are the advantages of using it?

Response 8: The referee raised an interesting question regarding the Gauss-Newton iterative method. It is used in our model selection to obtain a good initialization factor. (In subsection 2.8, paragraph 1, lines 340-347). As can be seen in (Van Eyndhoven et al. 2021), such technique makes the EEG-only CP decomposition very robust. Highly similar EEG signatures were found for almost all random initializations.

Point 9: Please discuss how to obtain the suitable parameter values used in the proposed solution.

Response 9: Suitable parameters are obtained by iteratively optimizing the cost function. To clarify how to obtain the suitable parameter values used in the proposed solution, we have added more details as follows:

  • We have added “cpd_nls with 2000 iterations, 400 conjugate gradient iterations for the step computation, and tolerance on the relative cost function update of , in Tensorlab 3.0” (In subsection 2.8, paragraph 1, lines 346-348)
  • We have added “to iteratively optimized Eq. (3) … sdf_minf with 1000 iterations, and tolerance on the relative cost function update of , in Tensorlab 3.0” (In subsection 2.8, paragraph 1, lines 350-352)

Point 10: What is the experimental environment?

Response 10: We have added the experimental environment in subsection 2.3, paragraph 2, lines 229-233 as follows:

“The experiments were conducted in the hospital's MRI scanning room. Electrode caps are MR-compatible, and the amplifier is placed in the MRI scanning room. A clock synchronization box was used to ensure the synchronicity of the EEG and fMRI data. During the experiment, the subject lay flat in the MR scanner, watched the gambling task on the screen and chose the door which he thought would win by pressing a button.”

Point 11: The experimental results are not convincing. Please the proposed solution with more recently published solutions.

Response 11: We also published a paper in Brain and Behavior based on this batch of data using EEG-informed fMRI analysis in 2017(Guo et al. 2017). We have added a comparison of CMTF method with the method in terms of reward processing results in section 4, paragraph 4, lines 556-560 as follows:

“In comparison with EEG-informed fMRI analysis of fusion results(Guo et al. 2017), the CMTF method showed stronger activation for both winning and losing stimuli in both insula and olfactory cortex. For winning stimuli, greater activation was shown in the supramarginal gyrus. In addition, activation of the inferior occipital gyrus and fusiform gyrus during winning and losing was found, possibly because they are associated with vision.”

We described the possible reasons for the activation of the three brain regions we newly identified with CMTF in section 4, paragraph 2, lines 521-538 as follows:

“For example, we found that the olfactory cortex, inferior occipital gyrus, and fusiform gyrus were also activated when winning and losing, which was not mentioned in other studies. The olfactory cortex is part of the limbic system which is involved in processing emotions, forming memories, and linking the senses to memories and emotions(McDonald 2014). This could be the reason why it was activated. In the meantime, the inferior occipital and fusiform gyrus were activated most likely because they are associated with vision - such as processing color information, face and body recognition, and word recognition(Weiner and Zilles 2016; Haas et al. 2021), which were all heavily involved in the gambling task utilized in this study.”

References

Chatzichristos C, Kofidis E, Van Paesschen W, De Lathauwer L, Theodoridis S, Van Huffel S. Early soft and flexible fusion of electroencephalography and functional magnetic resonance imaging via double coupled matrix tensor factorization for multisubject group analysis. Human Brain Mapping. 2022 Mar;43(4):1231–55.

Chatzichristos C, Vandecapelle M, Kofidis E, Theodoridis S, Lathauwer LD, Van Huffel S. Tensor-based Blind fMRI Source Separation Without the Gaussian Noise Assumption — A β-Divergence Approach. In: 2019 IEEE Global Conference on Signal and Information Processing (GlobalSIP). 2019. p. 1–5.

Dong G, Li H, Wang Y, Potenza MN. Individual differences in self-reported reward-approach tendencies relate to resting-state and reward-task-based fMRI measures. International Journal of Psychophysiology. 2018 Jun 1;128:31–9.

Guo Q, Zhou T, Li W, Dong L, Wang S, Zou L. Single-trial EEG-informed fMRI analysis of emotional decision problems in hot executive function. Brain and Behavior. 2017;7(7):e00728.

Haas B de, Sereno MI, Schwarzkopf DS. Inferior Occipital Gyrus Is Organized along Common Gradients of Spatial and Face-Part Selectivity. J Neurosci. 2021 Jun 23;41(25):5511–21.

Hunyadi B, Dupont P, Van Paesschen W, Van Huffel S. Tensor decompositions and data fusion in epileptic electroencephalography and functional magnetic resonance imaging data: Tensors in EEG-fMRI. WIREs Data Mining Knowl Discov. 2017 Jan;7(1):e1197.

Li X, Lu ZL, D’Argembeau A, Ng M, Bechara A. The Iowa Gambling Task in fMRI images. Human Brain Mapping. 2010;31(3):410–23.

McDonald AJ. Amygdala. In: Aminoff MJ, Daroff RB, editors. Encyclopedia of the Neurological Sciences (Second Edition) [Internet]. Oxford: Academic Press; 2014 [cited 2022 Sep 29]. p. 153–6. Available from: https://www.sciencedirect.com/science/article/pii/B9780123851574011131

Mosayebi R, Hossein-Zadeh GA. Correlated coupled matrix tensor factorization method for simultaneous EEG-fMRI data fusion. Biomedical Signal Processing and Control. 2020 Sep;62:102071.

Poldrack RA, Fletcher PC, Henson RN, Worsley KJ, Brett M, Nichols TE. Guidelines for reporting an fMRI study. NeuroImage. 2008 Apr 1;40(2):409–14.

Qin Y, Jiang S, Zhang Q, Dong L, Jia X, He H, et al. Ballistocardiogram artifact removal in simultaneous EEG-fMRI using generative adversarial network. NeuroImage: Clinical. 2019;22:101759.

Van Eyndhoven S, Dupont P, Tousseyn S, Vervliet N, Van Paesschen W, Van Huffel S, et al. Augmenting interictal mapping with neurovascular coupling biomarkers by structured factorization of epileptic EEG and fMRI data. NeuroImage. 2021 Mar;228:117652.

Weiner KS, Zilles K. The anatomical and functional specialization of the fusiform gyrus. Neuropsychologia. 2016 Mar 1;83:48–62.

Yazdi K, Rumetshofer T, Gnauer M, Csillag D, Rosenleitner J, Kleiser R. Neurobiological processes during the Cambridge gambling task. Behavioural Brain Research. 2019 Jan 1;356:295–304.

Reviewer 3 Report

Authors present a study on 20 healthy volunteers to investigate reward mechanisms while performing gambling task  to apply tensor decomposition to simultaneous electroencephalography (EEG) and functional magnetic resonance imaging (fMRI).EEG and fMRI data were jointly decomposed into a superposition of multiple sources characterized by space-time-frequency profiles using coupled matrix tensor factorization (CMTF) for neurovascular coupling. The orbitofrontal cortex, insula, and other regions were activated for both winning and losing stimuli; the superior orbital frontal gyrus, anterior cingulate, cortex, and other regions for the winning stimuli; the inferior frontal gyrus, cingulate cortex, medial superior frontal gyrus for the losing stimuli.

An MRI physicist/neuroradiologist is needed for evaluation of Materials and methods. 

I suggest to include current literature, cite and comment:

Warbrick T. Simultaneous EEG-fMRI: What Have We Learned and What Does the Future Hold? Sensors (Basel). 2022 Mar 15;22(6):2262. doi: 10.3390/s22062262. PMID: 35336434; PMCID: PMC8952790.   Gallego-Rudolf J, Corsi-Cabrera M, Concha L, Ricardo-Garcell J, Pasaye-Alcaraz E. Preservation of EEG spectral power features during simultaneous EEG-fMRI. Front Neurosci. 2022 Dec 23;16:951321. doi: 10.3389/fnins.2022.951321. PMID: 36620439; PMCID: PMC9816433.  

Dehghani A, Soltanian-Zadeh H, Hossein-Zadeh GA. Probing fMRI brain connectivity and activity changes during emotion regulation by EEG neurofeedback. Front Hum Neurosci. 2023 Jan 6;16:988890. doi: 10.3389/fnhum.2022.988890. PMID: 36684847; PMCID: PMC9853008.

Author Response

Responses to reviewers’ comments of manuscript ID brainsci-2267779

Reviewer: 3

Comments to the Author

Point: Authors present a study on 20 healthy volunteers to investigate reward mechanisms while performing gambling task to apply tensor decomposition to simultaneous electroencephalography (EEG) and functional magnetic resonance imaging (fMRI). EEG and fMRI data were jointly decomposed into a superposition of multiple sources characterized by space-time-frequency profiles using coupled matrix tensor factorization (CMTF) for neurovascular coupling. The orbitofrontal cortex, insula, and other regions were activated for both winning and losing stimuli; the superior orbital frontal gyrus, anterior cingulate, cortex, and other regions for the winning stimuli; the inferior frontal gyrus, cingulate cortex, medial superior frontal gyrus for the losing stimuli.

An MRI physicist/neuroradiologist is needed for evaluation of Materials and methods.

Response: Many thanks for the reviewer’s positive comments. We have revised the manuscript according to the reviewers’ suggestions.

Point 1: I suggest to include current literature, cite and comment:

Warbrick T. Simultaneous EEG-fMRI: What Have We Learned and What Does the Future Hold? Sensors (Basel). 2022 Mar 15;22(6):2262. doi: 10.3390/s22062262. PMID: 35336434; PMCID: PMC8952790.   Gallego-Rudolf J, Corsi-Cabrera M, Concha L, Ricardo-Garcell J, Pasaye-Alcaraz E. Preservation of EEG spectral power features during simultaneous EEG-fMRI. Front Neurosci. 2022 Dec 23;16:951321. doi: 10.3389/fnins.2022.951321. PMID: 36620439; PMCID: PMC9816433. 

Dehghani A, Soltanian-Zadeh H, Hossein-Zadeh GA. Probing fMRI brain connectivity and activity changes during emotion regulation by EEG neurofeedback. Front Hum Neurosci. 2023 Jan 6;16:988890. doi: 10.3389/fnhum.2022.988890. PMID: 36684847; PMCID: PMC9853008.

Response 1: Thank you very much for your kind review. We have added three references on the related work as follows:

  • “It is therefore of vital importance to find a way to fuse complementary different brain imaging techniques[12,13]” (In section 1, paragraph 3, lines 81-82)
  • “Dehghani et al.[22] implemented EEG neurofeedback with simultaneous fMRI to understand the effect of neurofeedback on brain activity and the interaction of whole brain regions involved in emotion regulation.” (In section 1, paragraph 3, lines 91-94)
  • “EEG data quality can be severely compromised when recording inside the MR environment[36], so denoising is necessary.” (In subsection 2.4, paragraph 1, lines 235-236)

References

Chatzichristos C, Kofidis E, Van Paesschen W, De Lathauwer L, Theodoridis S, Van Huffel S. Early soft and flexible fusion of electroencephalography and functional magnetic resonance imaging via double coupled matrix tensor factorization for multisubject group analysis. Human Brain Mapping. 2022 Mar;43(4):1231–55.

Chatzichristos C, Vandecapelle M, Kofidis E, Theodoridis S, Lathauwer LD, Van Huffel S. Tensor-based Blind fMRI Source Separation Without the Gaussian Noise Assumption — A β-Divergence Approach. In: 2019 IEEE Global Conference on Signal and Information Processing (GlobalSIP). 2019. p. 1–5.

Dong G, Li H, Wang Y, Potenza MN. Individual differences in self-reported reward-approach tendencies relate to resting-state and reward-task-based fMRI measures. International Journal of Psychophysiology. 2018 Jun 1;128:31–9.

Guo Q, Zhou T, Li W, Dong L, Wang S, Zou L. Single-trial EEG-informed fMRI analysis of emotional decision problems in hot executive function. Brain and Behavior. 2017;7(7):e00728.

Haas B de, Sereno MI, Schwarzkopf DS. Inferior Occipital Gyrus Is Organized along Common Gradients of Spatial and Face-Part Selectivity. J Neurosci. 2021 Jun 23;41(25):5511–21.

Hunyadi B, Dupont P, Van Paesschen W, Van Huffel S. Tensor decompositions and data fusion in epileptic electroencephalography and functional magnetic resonance imaging data: Tensors in EEG-fMRI. WIREs Data Mining Knowl Discov. 2017 Jan;7(1):e1197.

Li X, Lu ZL, D’Argembeau A, Ng M, Bechara A. The Iowa Gambling Task in fMRI images. Human Brain Mapping. 2010;31(3):410–23.

McDonald AJ. Amygdala. In: Aminoff MJ, Daroff RB, editors. Encyclopedia of the Neurological Sciences (Second Edition) [Internet]. Oxford: Academic Press; 2014 [cited 2022 Sep 29]. p. 153–6. Available from: https://www.sciencedirect.com/science/article/pii/B9780123851574011131

Mosayebi R, Hossein-Zadeh GA. Correlated coupled matrix tensor factorization method for simultaneous EEG-fMRI data fusion. Biomedical Signal Processing and Control. 2020 Sep;62:102071.

Poldrack RA, Fletcher PC, Henson RN, Worsley KJ, Brett M, Nichols TE. Guidelines for reporting an fMRI study. NeuroImage. 2008 Apr 1;40(2):409–14.

Qin Y, Jiang S, Zhang Q, Dong L, Jia X, He H, et al. Ballistocardiogram artifact removal in simultaneous EEG-fMRI using generative adversarial network. NeuroImage: Clinical. 2019;22:101759.

Van Eyndhoven S, Dupont P, Tousseyn S, Vervliet N, Van Paesschen W, Van Huffel S, et al. Augmenting interictal mapping with neurovascular coupling biomarkers by structured factorization of epileptic EEG and fMRI data. NeuroImage. 2021 Mar;228:117652.

Weiner KS, Zilles K. The anatomical and functional specialization of the fusiform gyrus. Neuropsychologia. 2016 Mar 1;83:48–62.

Yazdi K, Rumetshofer T, Gnauer M, Csillag D, Rosenleitner J, Kleiser R. Neurobiological processes during the Cambridge gambling task. Behavioural Brain Research. 2019 Jan 1;356:295–304.

Round 2

Reviewer 1 Report

The authors have addressed all my concerns well.

Reviewer 2 Report

All my concerns have been addressed. 

Reviewer 3 Report

Authors have responded to my remarks, I suggest to include an MRI physicist/neuroradiologist for final evaluation of the described methods.